# Metabolomic Analysis Identifies Betaine as a Key Mediator of *TAp73α*-Induced Ferroptosis in Ovarian Granulosa Cells

**DOI:** 10.3390/ijms26136045

**Published:** 2025-06-24

**Authors:** Liping Mei, Le Chen, Bingfei Zhang, Xianbo Jia, Xiang Gan, Wenqiang Sun

**Affiliations:** 1Key Laboratory of Livestock and Poultry Multi-Omics, Ministry of Agriculture and Rural Affairs, College of Animal Science and Technology, Sichuan Agricultural University, Chengdu 611130, China; 15700312879@163.com (L.M.); 15990973537@163.com (L.C.); zhangbingfei07@163.com (B.Z.); jaxb369@sicau.edu.cn (X.J.); 2Scientific Research Center, Guilin Medical University, Guilin 541199, China; ganxiangdk@163.com

**Keywords:** granulosa cells, cattle, *TAp73α*, metabolomics, ferroptosis, betaine

## Abstract

Granulosa cells (GCs) are essential for follicular growth and development, and their functional state critically impacts folliculogenesis. *TAp73α*, a transcriptionally active isoform of the *p73* gene, is crucial for maintaining follicular integrity. In this study, we demonstrate that *TAp73α* overexpression promotes ferroptosis in bovine GCs by downregulating *SLC7A11*, depleting intracellular glutathione (GSH), and enhancing lipid peroxidation, particularly under Erastin treatment. By contrast, *TAp73α* knockdown restores antioxidant capacity, elevates GSH levels, and attenuates ferroptosis. To elucidate the underlying mechanism, untargeted metabolomic profiling revealed that *TAp73α* overexpression significantly altered the metabolic landscape of GCs, with marked enrichment in the glutathione metabolism pathway. Notably, betaine—a metabolite closely linked to redox homeostasis—was markedly downregulated. Functional assays confirmed that exogenous betaine supplementation restored *SLC7A11* expression, increased GSH levels, and alleviated oxidative damage induced by either H_2_O_2_ or *TAp73α* overexpression. Moreover, betaine co-treatment effectively reversed lipid peroxide accumulation and mitigated *TAp73α*-induced ferroptosis. Collectively, our findings identify a novel mechanism by which *TAp73α* promotes ferroptosis in granulosa cells through the suppression of betaine and glutathione metabolism, highlighting betaine as a key metabolic modulator with promising protective potential.

## 1. Introduction

As the core functional unit of the follicular microenvironment, granulosa cells (GCs) play a crucial role in regulating follicular development, maturation, and the maintenance of female reproductive function [1]. GCs not only contribute to ovarian endocrine homeostasis by secreting steroid hormones such as estrogen and progesterone but also maintain follicular structural integrity through bidirectional signaling with the oocyte [2]. Thus, GCs are essential for facilitating follicular growth.

The *p73* gene, first identified in 1997, undergoes alternative splicing at both its 3′ end (generating α, β, γ, and other isoforms) and 5′ end (yielding *TAp73* and *ΔNp73* isoforms), resulting in at least 14 distinct transcript variants. Among these, *TAp73α* is the most abundantly expressed isoform [3]. As a member of the *p53* gene family, *p73* functions as a transcription factor involved in regulating the cell cycle, inducing apoptosis, and maintaining genomic stability. Studies have shown that female mice deficient in *TAp73* exhibit abnormal meiotic spindle formation during oocyte maturation, leading to ovulatory defects [4]. During follicular maturation, *p73* regulates the expression of granulosa cell-specific hormone receptors, enhancing their sensitivity to gonadotropins such as FSH and LH, thereby promoting follicular development [5]. Moreover, *p73*-deficient mice display dysfunction of the hypothalamic–pituitary–gonadal (HPG) axis, including hippocampal hypoplasia and hydrocephalus [6]. These abnormalities may disrupt the normal secretion of gonadotropin-releasing hormone (GnRH), ultimately impairing the secretion of LH and FSH [7]. Collectively, these findings highlight the critical role of *p73* in coordinating follicular development and reproductive endocrine function.

During different stages of follicular development, granulosa cells undergo complex degenerative processes, among which apoptosis and autophagy are recognized as key molecular mechanisms governing cell fate [8,9]. Programmed cell death pathways, such as those regulated by the Bcl-2 family and caspase cascades, are major contributors to follicular atresia and are implicated in pathological conditions including premature ovarian failure [10]. In addition to apoptosis, recent studies have identified ferroptosis in ovarian granulosa cells across multiple species, including pigs, mice, and sheep [11,12,13]. However, the mechanisms underlying ferroptosis in granulosa cells remain poorly understood and require further systematic investigation.

Studies have shown that in cortical neurons of *TAp73* knockout mice, levels of glucose and metabolites related to glycolysis were elevated, suggesting that *TAp73* may act as a negative regulator of glycolysis [14] In Saos-2 cells, overexpression of *TAp73* significantly enhanced mitochondrial activity and activated the pentose phosphate pathway (PPP), indicating that *p73* can influence cellular function through metabolic reprogramming [15]. Furthermore, deletion of exon 13 in the *p73* gene, which promotes a shift from the *p73*α to *p73*β isoform, led to increased ferroptosis [16]. This phenotype could be rescued by silencing *p73β*, suggesting that *p73* may play a role in regulating ferroptosis, potentially through modulation of *CDO1*. Ferroptosis is a form of iron-dependent programmed cell death characterized by the accumulation of lipid peroxides and depletion of GSH [17]. As summarized in Figure 1, we propose a conceptual model in which ferroptosis is regulated through pathways involving *p73 SLC7A11, GPX4*, GSH, ROS, and Nrf2. However, whether *p73* regulates ferroptosis through its broader impact on cellular metabolism remains unclear and warrants further investigation.

Therefore, in this study, we investigated the regulatory effect of *TAp73α* on ferroptosis in bovine granulosa cells and performed untargeted metabolomics to identify key metabolites involved in this process. Notably, our metabolomic analysis revealed that TAp73α downregulates betaine, a key antioxidant-associated metabolite implicated in ferroptosis regulation. Our findings provide novel insights into the metabolic regulation of ferroptosis in granulosa cells and highlight betaine as a potential protective modulator against *TAp73α*-induced oxidative stress.

## 2. Results

### 2.1. TAp73α Overexpression Promotes Ferroptosis in Granulosa Cells

Extensive evidence has shown that granulosa cells are susceptible to ferroptosis [11,12,13]. In this study, bovine GCs were treated with Erastin (C_30_H_31_ClN_4_O_4_), a classical ferroptosis inducer, and a dose-dependent increase in lipid peroxidation was observed, confirming the cells’ sensitivity to ferroptosis (Appendix A). Previous studies have confirmed that *p73* is specifically expressed in the granulosa layer of follicles and plays a critical role in follicular development [5]. To further investigate the role of *p73*, we constructed a *TAp73α* overexpression plasmid and transfected it into GCs. Following transfection, *TAp73α* expression was significantly upregulated (Figure 2A), accompanied by increased lipid peroxide accumulation (Figure 2B). Co-treatment with Erastin further exacerbated ferroptosis (Figure 2C). In line with metabolomic findings indicating involvement of the glutathione pathway, we found that *TAp73α* overexpression markedly suppressed the expression of *SLC7A11* (Figure 2D), resulting in reduced GSH levels (Figure 2E), with an additional drop under Erastin treatment (Figure 2F). These results suggest that *TAp73α* may promote ferroptosis in GCs by inhibiting GSH synthesis.

### 2.2. TAp73α Knockdown Attenuates Ferroptosis in Granulosa Cells

To further validate the role of *TAp73α* in ferroptosis, we knocked down *TAp73α* in bovine GCs (Figure 3A) and treated the cells with Erastin. Knockdown of *TAp73α* significantly alleviated Erastin-induced ferroptosis (Figure 3B). Additionally, expression of *SLC7A11* was restored (Figure 3C), and GSH levels were significantly increased (Figure 3D). A further rise in GSH content was observed under co-treatment with Erastin (Figure 3E). Collectively, these findings suggest that *TAp73α* regulates ferroptosis in granulosa cells by modulating glutathione biosynthesis.

### 2.3. Effect of TAp73α Overexpression on the Metabolomics of GCs in Positive Mode

To further explore the molecular mechanisms by which *TAp73α* regulates GCs, we overexpressed *TAp73α* in GCs and conducted untargeted metabolomic profiling. In positive ion mode, a total of 636 metabolites were significantly altered in the *TAp73α*-overexpressing group compared to the control group, with 495 downregulated and 141 upregulated metabolites (Figure 4A, Appendix A). Clustering analysis showed distinct separation between the two groups (Figure 4B). The most upregulated metabolites included (1r, 2s)-2-aminocyclopentane carboxylic acid, N’-hydroxysaxitoxin, and 4-hydroxyandrostenedione glucuronide, while the most downregulated were indoxyl glucuronide, 4,5-dimethyl-2-propyloxazole, and 8-methoxykynurenate (Figure 4C). Pathway enrichment analysis revealed that these differential metabolites were primarily enriched in the arachidonic acid metabolism and glutathione metabolism pathways (Figure 4D), both of which are closely associated with ferroptosis.

### 2.4. Effect of TAp73α Overexpression on the Metabolomics of GCs in Negative Mode

In negative ion mode, 723 metabolites were significantly altered in the *TAp73α*-overexpressing group, including 446 downregulated and 277 upregulated metabolites (Figure 5A, Appendix A). Similar to the positive ion mode, clustering analysis revealed a clear separation between the groups (Figure 5B). The top upregulated metabolites were neosaxitoxin, Met-Asp, and pravastatin, while the most downregulated included CMP-5-N-acetyl-7-N-(D-alanyl)-legionaminic acid, puromycin, and lysyltyrosine (Figure 5C). Pathway enrichment analysis further confirmed significant enrichment of differential metabolites in arachidonic acid metabolism and glutathione metabolism (Figure 5D), reinforcing the potential role of *TAp73α* in the regulation of ferroptosis in granulosa cells.

### 2.5. Betaine Enhances Antioxidant Capacity of GCs

Metabolomic analysis revealed a significant reduction in betaine levels following *p73* overexpression (Appendix A). In recent years, numerous studies have demonstrated that betaine enhances cellular antioxidant defenses and protects ovarian tissue from oxidative damage [18]. These findings suggest that *p73* may modulate GC function through the regulation of betaine metabolism. To investigate the effect of betaine on GCs, we treated the cells with a range of betaine concentrations. CCK-8 assays showed that cell viability initially increased and then decreased, reaching its peak at 5 mM (Figure 6A). At this concentration, betaine significantly upregulated *SLC7A11* mRNA expression (Figure 6B) and markedly increased intracellular GSH levels (Figure 6C). Furthermore, co-treatment with H_2_O_2_ and betaine demonstrated that betaine substantially alleviated H_2_O_2_-induced cytotoxicity (Figure 6D), providing additional evidence that betaine enhances the antioxidant capacity of GCs.

### 2.6. Betaine as a Key Metabolite in the Modulation of Ferroptosis by TAp73α

To investigate the role of betaine in regulating ferroptosis in granulosa cells (GCs), we first co-treated GCs with betaine and Erastin. The results showed that betaine significantly attenuated the Erastin-induced depletion of intracellular GSH (Figure 7A) and reduced lipid peroxidation accumulation caused by Erastin (Figure 7B). To further determine whether betaine is a key mediator in *p73*-regulated ferroptosis, we co-treated GCs with betaine and *TAp73α* overexpression. Betaine significantly reversed the inhibitory effect of *TAp73α* on *SLC7A11* mRNA expression (Figure 7C) and alleviated the *TAp73α*-induced accumulation of lipid peroxides in GCs (Figure 7D). Collectively, these findings suggest that betaine is a critical metabolite involved in *p73*-mediated regulation of ferroptosis in granulosa cells.

## 3. Discussion

Granulosa cells are essential for follicular growth, maturation, and the maintenance of female reproductive function. As the central functional units within the follicle, granulosa cells play critical roles throughout the dynamic processes of follicular recruitment, selection, and dominance, with their fate decisions directly influencing the progression of follicular development [19]. Numerous studies have demonstrated that the physiological activities of granulosa cells, such as proliferation and apoptosis, are tightly regulated by a complex network of genes and signaling pathways [10,20]. Therefore, elucidating the molecular mechanisms governing granulosa cell development and function is crucial for understanding the overall process of folliculogenesis.

The *p73* gene is transcribed from two distinct promoters, giving rise to two major isoform groups: the full-length *TAp73* and the N-terminal truncated ΔN*p73* [3]. In male mice, *p73* regulates the adhesion between developing spermatids and Sertoli cells; loss of TAp73 results in premature germ cell detachment and apoptosis, resulting in severe sperm depletion and infertility [21]. In the ovary, granulosa cells surround and support the oocyte, playing a central role in coordinating follicle development and oocyte maturation. The *p73* gene is expressed in ovarian granulosa cells, where it participates in precise transcriptional regulation essential for cellular homeostasis and differentiation [5,22]. Taken together, these findings indicate that *p73* is essential for maintaining normal follicular development and female reproductive competence.

To validate this hypothesis, we treated bovine granulosa cells with Erastin, a classical ferroptosis inducer, and observed a dose-dependent increase in lipid peroxidation, indicating a high sensitivity of granulosa cells to ferroptosis. Overexpression of *TAp73α* significantly promoted lipid peroxide accumulation and further enhanced ferroptotic responses when combined with Erastin treatment. Mechanistically, *TAp73α* overexpression downregulated *SLC7A11* expression and reduced intracellular GSH levels, suggesting that it promotes ferroptosis by impairing glutathione biosynthesis. Conversely, knockdown of *TAp73α* alleviated Erastin-induced ferroptosis, restored *SLC7A11* expression, and increased GSH levels.

Studies have shown that *TAp73* may reprogram glucose metabolism by transcriptionally activating PFKL (a rate-limiting enzyme in glycolysis) and G6PD (a key enzyme in the pentose phosphate pathway), thereby providing energy for cellular processes and maintaining redox homeostasis via NADPH production [23]. In addition, *TAp73* interacts with signaling pathways such as PI3K/Akt, modulating Akt phosphorylation to counteract anti-apoptotic signals [24]. *TAp73* also plays a critical role in the serine/glycine biosynthesis pathway. Metabolic profiling of human cancer cells revealed that *TAp73* transcriptionally regulates glutaminase-2 (GLS-2), thereby influencing serine biosynthesis [25]. These findings collectively suggest that *p73* exerts its biological functions, at least in part, through the transcriptional control of cellular metabolic pathways.

In addition to glutathione metabolism, differential metabolites were also enriched in fatty acid metabolism pathways, notably the arachidonic acid pathway. Two bioactive lipid mediators, prostaglandin J2 (PGJ2) and 12-oxoETE, derived from arachidonic acid metabolism, were significantly enriched, suggesting their potential role in ferroptosis regulation. Among them, 15deoxy-Δ^12,14^-prostaglandin J2 (15d-PGJ2), a downstream metabolite of PGJ2, is known to activate the transcription factor Nrf2 [26]. Nrf2 plays a pivotal role in regulating anti-ferroptotic genes, the downstream targets of which inhibit lipid peroxidation and reduce intracellular free iron accumulation [27]. Moreover, deletion of exon 13 in *p73*, which induces a shift from the *p73*α to *p73β* isoform, was associated with increased ferroptosis. This phenotype could be reversed by silencing *p73β* [16], further supporting a regulatory role for *p73* in ferroptosis.

To investigate the molecular mechanisms by which *p73* regulates ovarian granulosa cells, we performed metabolomic profiling of bovine granulosa cells overexpressing *TAp73α*. In positive ion mode, a total of 636 significantly altered metabolites were identified compared to the control group, including 495 downregulated and 141 upregulated metabolites. In negative ion mode, 723 differential metabolites were detected, with 446 downregulated and 277 upregulated. Pathway enrichment analysis of these differential metabolites revealed a predominant involvement of the glutathione metabolism pathway. GSH, a tripeptide composed of glutamate (Glu), cysteine (Cys), and glycine (Gly), is a major intracellular antioxidant. A reduction in GSH levels reflects compromised cellular antioxidant capacity [28]. Ferroptosis, a type of iron-dependent programmed cell death, is characterized by excessive lipid peroxidation and GSH depletion [17]. GSH plays a crucial role in defending against oxidative stress and suppressing ferroptosis. Its synthesis and degradation directly affect cellular sensitivity to ferroptosis [29]. During ferroptosis, GSH depletion and excessive lipid peroxidation amplify oxidative stress, forming a positive feedback loop that accelerates cell death.

Notably, metabolomic analysis revealed a significant decrease in betaine levels following *p73* overexpression. Betaine (C_5_H_11_NO_2_) is a naturally occurring quaternary ammonium compound derived from glycine with three methyl groups. It functions as a methyl donor in various metabolic pathways and is widely distributed in plants and animals. Betaine also acts as an osmoprotectant and antioxidant, contributing to cellular homeostasis and stress resistance. Accumulating evidence indicates that betaine plays important roles in reproductive function and oxidative stress resistance. For instance, betaine supplementation delays hyperglycemia-induced aging of testicular and ovarian cells [30], elevates serum luteinizing hormone (LH) and progesterone levels, shortens the weaning-to-estrus interval, and increases litter size in sows [31]. Betaine also regulates reproductive hormone secretion [32]. In laying hens, dietary betaine (0.05–0.15%) enhances egg production and increases vitellogenin (VTG) levels [33]. Betaine has also been shown to exert strong antioxidant effects by directly scavenging reactive oxygen species (ROS), thereby reducing oxidative damage to cells and biomolecules. For example, betaine alleviates oxidative stress in MAC-T cells by inhibiting ROS accumulation [34] and reverses heat stress-induced increases in testicular malondialdehyde (MDA) levels. In broilers, dietary betaine significantly reduces MDA content in breast and thigh muscles and enhances glutathione peroxidase (GPX) activity and intracellular GSH levels, indicating its ability to counteract oxidative damage [35]. In our study, CCK-8 assays showed that betaine promoted GC viability in a concentration-dependent manner, with the highest activity observed at 5 mM. At this concentration, betaine significantly upregulated *SLC7A11* mRNA expression and increased intracellular GSH levels. Co-treatment with H_2_O_2_ and betaine significantly attenuated H_2_O_2_-induced cytotoxicity, further confirming betaine’s antioxidant effects in GCs. Additionally, betaine alleviated Erastin-induced GSH depletion and reduced lipid peroxidation. To further determine whether betaine is a key metabolite involved in *p73*-mediated ferroptosis, GCs were co-treated with betaine and *TAp73α* overexpression. Betaine significantly reversed the *TAp73α*-induced suppression of *SLC7A11* and reduced lipid peroxide accumulation in GCs. Collectively, these findings suggest that betaine is a critical downstream metabolite in the p73 regulatory network, capable of mitigating ferroptosis by restoring the antioxidant defense system in granulosa cells. A schematic representation of this proposed mechanism is illustrated in Figure 8.

## 4. Materials and Methods

### 4.1. Cell Culture

Bovine ovaries were collected and disinfected by immersion in 75% ethanol for 30 s, followed by 3–5 washes with sterile phosphate-buffered saline (PBS). Excess connective tissue and fat were removed using sterile ophthalmic scissors. Follicles were dissected and mechanically punctured using a sterile 1 mL needle. Granulosa cells were released by gently scraping the inner follicular wall and cortical surface with the blunt side of sterile forceps. The cell suspension was collected into dishes containing culture medium. The cell suspension was centrifuged at 1000 rpm for 3 min. The pellet was resuspended in fresh complete medium and seeded into T25 culture flasks. Once the cells reached 85–90% confluence, the cells were seeded into 6-well plates and allowed to adhere for 24 h prior to further treatment. For ferroptosis induction, GCs were treated with varying concentrations of Erastin (0–2 μM) for 24 h. For betaine treatment, GCs were treated with varying concentrations of betaine (0–15 mM) for 24 h. Following treatment, indicators such as lipid peroxidation and intracellular GSH levels were assessed.

### 4.2. Cell Transfection

To achieve *TAp73α* overexpression, the full-length bovine *TAp73α* cDNA synthesized by (TSINGKE, Chengdu, China) was subcloned into the pcDNA3.1 vector (Invitrogen, Shanghai, China) to construct the mammalian expression plasmid pcDNA3.1-*TAp73α*. The empty pcDNA3.1 vector served as the negative control. Plasmid transfection was performed using Lipofectamine 3000 (Invitrogen, Carlsbad, CA, USA).

### 4.3. Metabolite Extraction

Metabolomics analysis was performed using a Waters Acquity I-Class PLUS ultra-high performance liquid chromatography system coupled with a Waters Xevo G2-XS QTof high-resolution mass spectrometer (Waters Corporation in Milford, MA, USA). Metabolite separation was achieved using a Waters Acquity UPLC HSS T3 column. Both positive and negative ion modes were employed, with mobile phase A consisting of 0.1% formic acid in water and mobile phase B consisting of 0.1% formic acid in acetonitrile. The injection volume for each sample was 2 μL.

### 4.4. LC-MS/MS Analysis

Mass spectrometric data acquisition was conducted in MSe mode using MassLynx V4.2 software (Waters, Milford, MA, USA), allowing simultaneous collection of precursor and product ion data under low and high collision energies within a single scan cycle. The low collision energy setting was turned off, while the high collision energy ranged from 10 to 40 V. Each mass spectrum was acquired with a scan time of 0.2 s. The electrospray ionization (ESI) source parameters were as follows: capillary voltage of +2500 V (positive mode) or −2000 V (negative mode), cone voltage of 30 V, source temperature of 100 °C, desolvation temperature of 500 °C, cone gas flow rate of 50 L/h, and desolvation gas flow rate of 800 L/h.

### 4.5. Data Preprocessing and Annotation

Raw data files obtained from LC-MS/MS analysis were processed using Progenesis QI software (version 4.0). This included steps such as peak picking, alignment, and normalization. Metabolite identification was conducted by comparing mass spectrometry data against entries in the METLIN online database and a custom-built in-house metabolite library, both integrated into the Progenesis QI software platform.

Normalized peak area data were subjected to multivariate and statistical analyses. Principal component analysis (PCA) and Spearman correlation analysis were used to evaluate intra-group sample consistency and quality control performance. Identified metabolites were annotated by referencing KEGG, HMDB, and LipidMaps databases to obtain classification and pathway information [36,37]. Fold change analysis and Student’s *t*-test were used to determine significantly different metabolites (*p* < 0.05). OPLS-DA modeling was carried out using the R package “ropls,” (Version 1.6.2) and model reliability was validated with 200 permutation tests. Variable importance in projection (VIP) scores were calculated through multiple cross-validation. Differential metabolites were screened based on criteria of FC > 1, *p* < 0.05, and VIP > 1. KEGG pathway enrichment of these differential metabolites was evaluated using hypergeometric distribution testing.

### 4.6. GSH Determination

GSH levels were determined using the Micro Reduced Glutathione (GSH) Assay Kit (Manufacturer: Solarbio Life Sciences (Beijing, China), Catalog Number: BC1175) according to the manufacturer’s instructions. Briefly, after treatment, the cells were collected by trypsinization. The cells were first washed with PBS, then lysed by ultrasonic disruption in lysis buffer. The lysates were centrifuged at 12,000× *g* for 10 min at 4 °C, and the supernatants were collected. In a 96-well plate, standards, sample supernatants, and blank controls were prepared. GSH content was measured based on a kinetic enzymatic recycling method that detects the oxidation of GSH by 5,5’-dithiobis-(2-nitrobenzoic acid) (DTNB) and glutathione reductase. The absorbance was recorded at 412 nm.

### 4.7. Detection of Lipid Peroxidation

The lipid peroxidation content was determined using the Lipid Peroxidation Assay Kit with BODIPY 581/591 C11 (Manufacturer: Beyotime Institute of Biotechnology (Shanghai, China), Catalog Number: S0043S) according to the manufacturer’s instructions. Bovine granulosa cells from different treatment groups were cultured in 12-well plates until adherence. After discarding the culture medium, the cells were gently washed twice with PBS to remove residual serum. BODIPY 581/591 C11 probe was then added to each well, and the plate was gently shaken to ensure even coverage of the cells. The cells were incubated in the dark for 30 min at 37 °C in a 5% CO_2_ incubator. Following incubation, the probe solution was removed, and each well was washed twice with PBS to eliminate unbound probes and cellular debris. Fluorescent images were acquired under a fluorescence microscope, capturing both green fluorescence (oxidized probe) and red fluorescence (non-oxidized probe). The degree of lipid peroxidation was quantified by calculating the ratio of green to red fluorescence.

### 4.8. ROS Determination

ROS levels were determined using the Reactive Oxygen Species Assay Kit (Solarbio Life Sciences) according to the manufacturer’s instructions. Bovine granulosa cells were seeded in 12-well plates and cultured at 37 °C with 5% CO_2_. After adherence and treatment, the cells were washed twice with PBS to remove residual serum. DCFH-DA probe was then added, and the cells were incubated at 37 °C in the dark for 20 min to detect intracellular ROS. After incubation, the cells were washed twice with PBS to remove excess probe and debris. Subsequently, DAPI staining was performed by incubating the cells at 37 °C in the dark for 8 min, followed by 2–3 PBS washes to eliminate unbound dye. Fluorescence microscopy was used to capture DCFH-DA (green) and DAPI (blue) signals.

### 4.9. CCK-8 Assay

Cell viability was determined using the Cell Counting Kit-8 (CCK-8) Assay with the CCK-8 Kit (Coolaber) according to the manufacturer’s instructions. Bovine granulosa cells were seeded into 96-well plates and divided into blank, control, and treatment groups. After the designated treatments, CCK-8 solution was added to each well, followed by gentle shaking to ensure thorough mixing with the culture medium. The plate was then incubated at 37 °C with 5% CO_2_ for an additional 2 h. Absorbance at 450 nm was measured using a microplate reader (Thermo Scientific, Waltham, MA, USA) to determine the optical density (OD) of each well.

### 4.10. Real-Time Quantitative PCR Analysis

Total RNA was isolated from cells using TRIzol reagent (Aidlab, Beijing, China), and its concentration and purity were assessed using a NanoDrop 2000 spectrophotometer (Thermo Scientific, Wilmington, DE, USA). Genomic DNA was removed by DNase I treatment, after which cDNA was synthesized using the PrimeScript RT Master Mix kit (Vazyme, Nanjing, China). Quantitative real-time PCR (RT-qPCR) was performed using the SYBR Premix Ex Taq™ (Takara, Dalian, China). The thermal cycling conditions were as follows: initial denaturation at 95 °C for 10 min, followed by 40 cycles of denaturation at 95 °C for 10 s and annealing/extension at 60 °C for 40 s. Fluorescence data were collected at 60 °C. Primer sequences are provided in Table 1. Relative quantification was performed using the 2^−ΔΔCt^ method [38].

### 4.11. Statistical Analysis

All data are expressed as the mean ± standard deviation (SD). Comparisons between two groups were performed using Student’s *t*-test, while differences among multiple groups were analyzed by one-way analysis of variance (ANOVA) followed by Tukey’s post hoc test.

## 5. Conclusions

In summary, our study highlights betaine not only as a critical downstream metabolic effector of TAp73α but also as a potential protective modulator against ferroptosis in granulosa cells. These findings provide new insights into the metabolic regulation of redox homeostasis in ovarian function and suggest that betaine may serve as a promising therapeutic target for preventing ferroptosis-associated reproductive disorders. Future studies should aim to validate these findings in vivo using animal models to evaluate the protective effects of dietary or pharmacological betaine supplementation on ovarian function and fertility outcomes. Additionally, investigating the upstream regulatory mechanisms of TAp73α expression under both physiological and pathological conditions may uncover novel signaling pathways involved in follicular atresia.

## Figures and Tables

**Figure 1 ijms-26-06045-f001:**
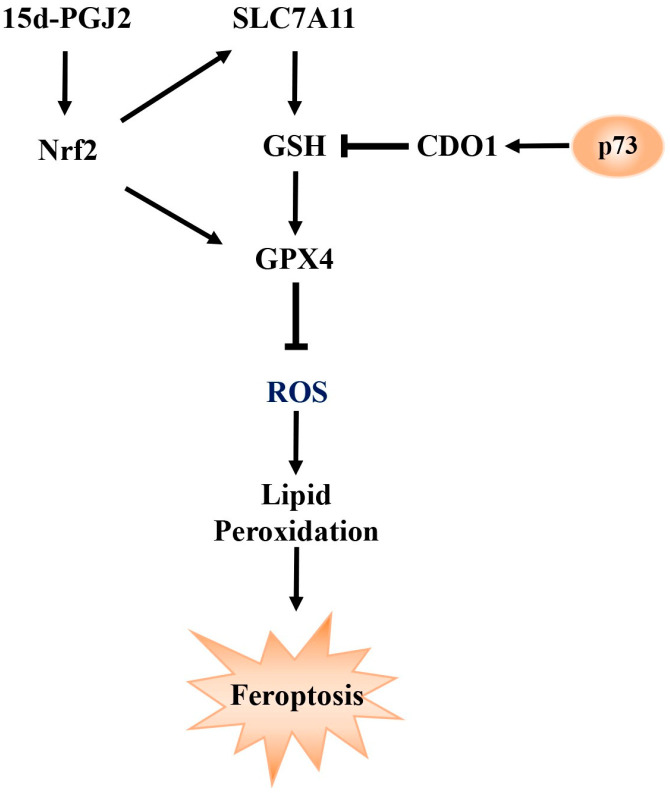
**A Conceptual Model of Ferroptosis Regulation Involving** *p73***, *SLC7A11*, *GPX4*, GSH, ROS, and Nrf2** *SLC7A11* facilitates the synthesis of GSH, which, under the catalytic action of *GPX4*, reduces intracellular reactive oxygen species (ROS) and lipid peroxidation levels, thereby suppressing ferroptosis. In addition, 15-deoxy-Δ^12,14^-prostaglandin J2 (15d-PGJ2) activates the Nrf2 signaling pathway, leading to the transcriptional upregulation of both *SLC7A11* and *GPX4*, further strengthening the cellular antioxidant defense and resistance to ferroptosis. Moreover, *p73* has been shown to modulate ferroptosis by regulating the expression of *CDO1,* a key gene involved in cysteine metabolism and redox balance. Arrowheads represent activation; blunt ends represent inhibition.

**Figure 2 ijms-26-06045-f002:**
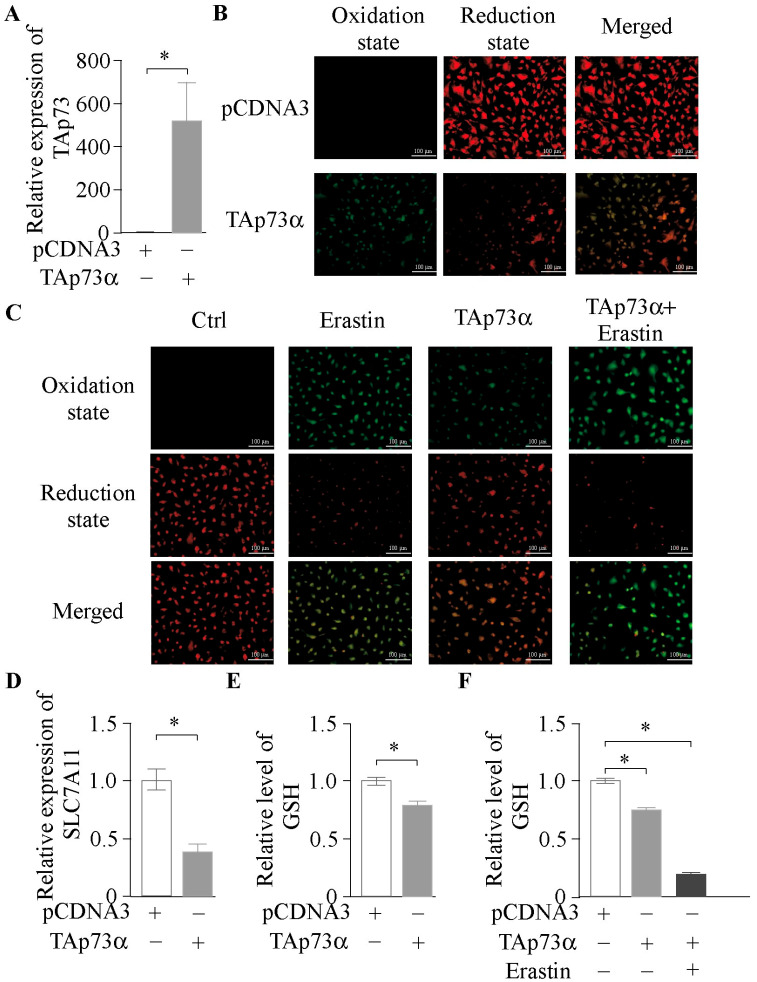
***TAp73α* overexpression enhances lipid peroxidation and suppresses glutathione metabolism in GCs.** (**A**) Relative expression levels of *TAp73* in follicular granulosa cells following *TAp73α* plasmid transfection. (**B**) Lipid ROS level after *TAp73α* plasmid transfection. (**C**) Lipid ROS level after Erastin, *TAp73α* plasmid, or their combination treatment. (**D**) Relative expression levels of *SLC7A11* in follicular granulosa cells following *TAp73α* plasmid transfection. (**E**) GSH level after *TAp73α* plasmid transfection. (**F**) GSH level after Erastin, *TAp73α* plasmid, or their combination treatment. Data are presented as mean ± SD. * *p* < 0.05.

**Figure 3 ijms-26-06045-f003:**
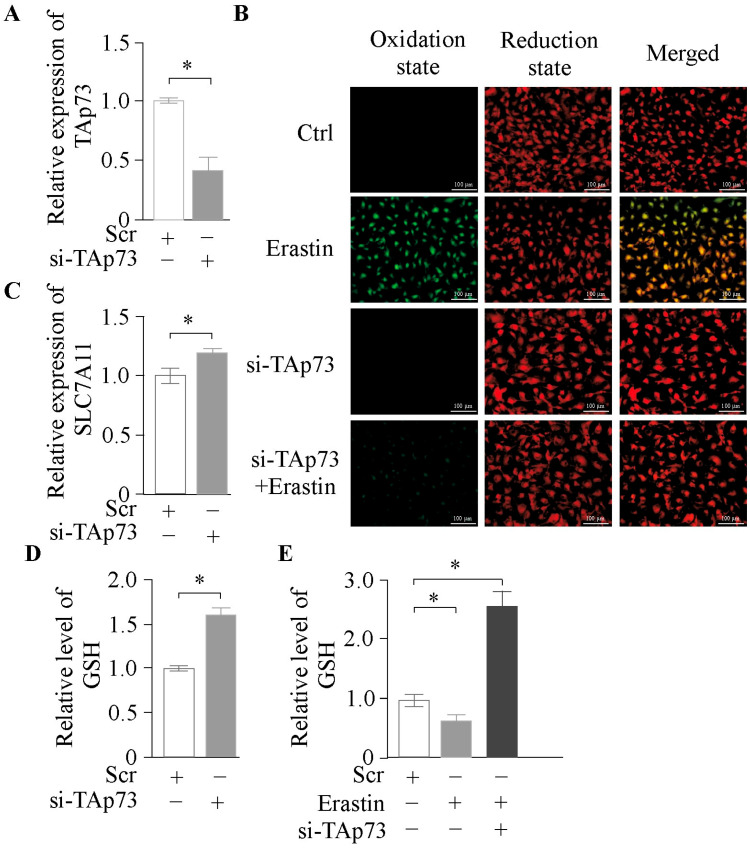
***TAp73* knockdown suppresses lipid peroxidation and enhances glutathione metabolism in GCs.** (**A**) Relative expression levels of *TAp73* in follicular granulosa cells following si-*TAp73* transfection. (**B**) Lipid ROS level after Erastin, si-*TAp73*, or their combination treatment. (**C**) Relative expression levels of *SLC7A11* in follicular granulosa cells following si-*TAp73* transfection. (**D**) GSH level after si-*TAp73* transfection. (**E**) GSH level after Erastin, si-*TAp73*, or their combination treatment. Data are presented as mean ± SD. * *p* < 0.05.

**Figure 4 ijms-26-06045-f004:**
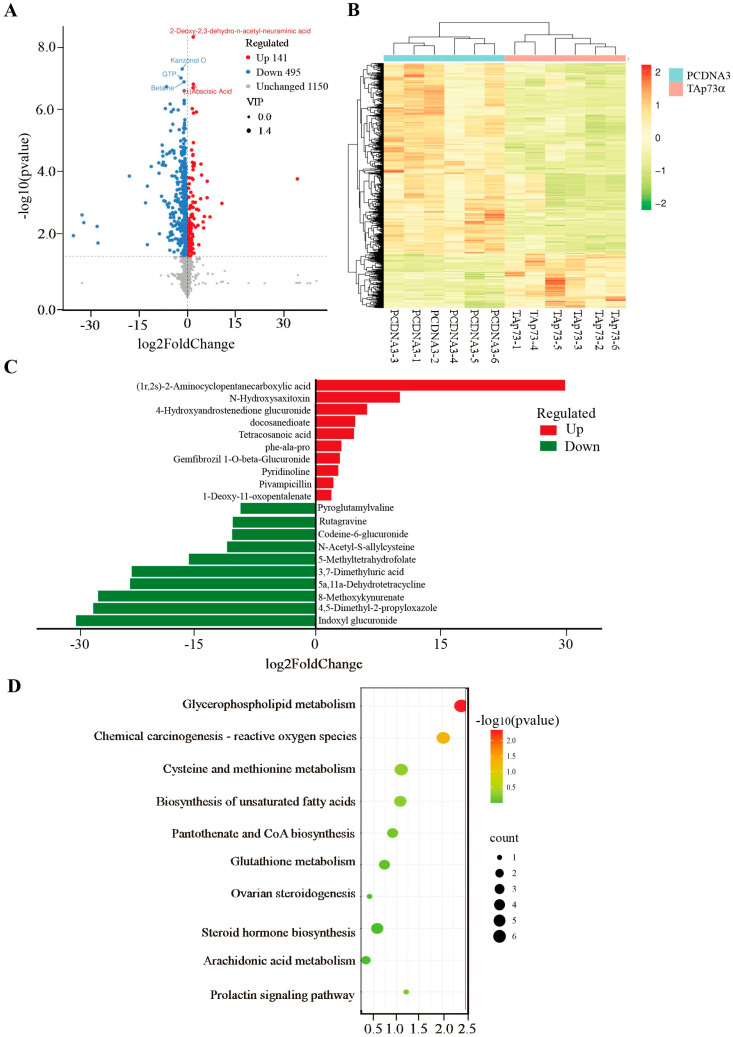
**Effect of *TAp73α* overexpression on the metabolomics of GCs in a positive model.** (**A**) Volcano plot of metabolites with significant differences between control and *TAp73α* overexpression groups. The criteria for upregulated differential metabolites were VIP > 1, *p* < 0.05, and log2 (fold change) > 0, while the criteria for downregulated differential metabolites were VIP > 1, *p* < 0.05, and log2 (fold change) < 0. (**B**) Clustering heat map of differential metabolites between control and *TAp73α* overexpression groups. (**C**) Top 10 elevated and reduced differential metabolites identified between the control and *TAp73α* overexpression groups in the positive model. (**D**) KEGG pathways enriched by differential metabolites between control and *TAp73α* overexpression groups. The dashed line represents -log10(*p*-value), corresponding to *p* < 0.05.

**Figure 5 ijms-26-06045-f005:**
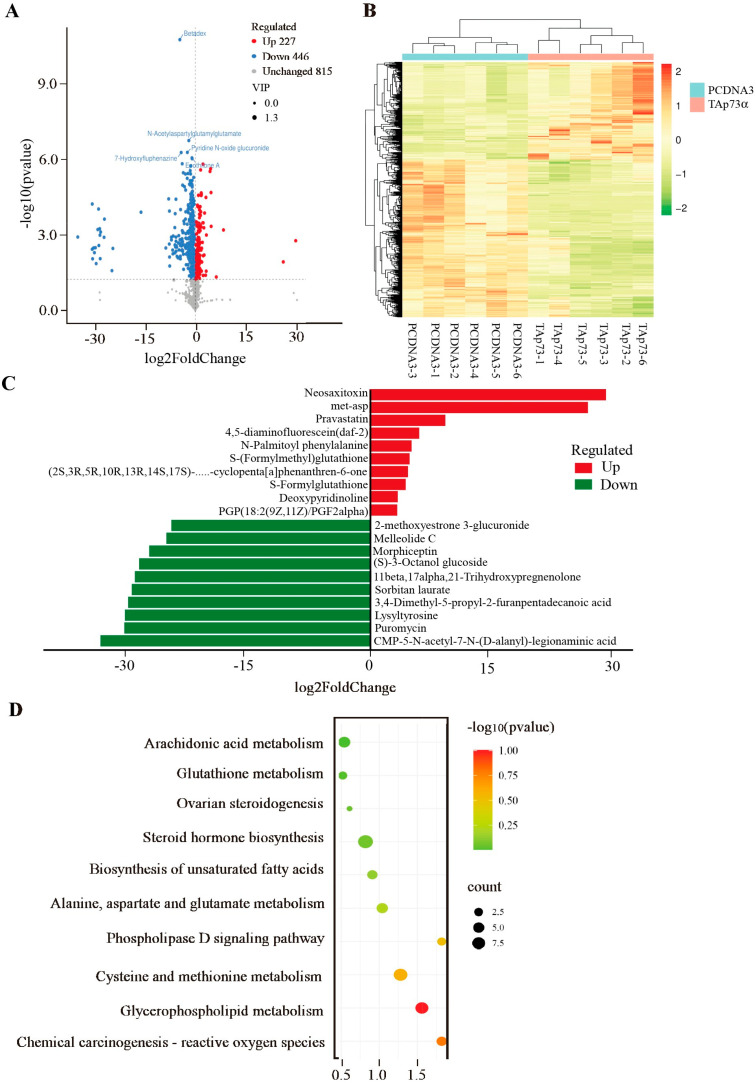
**Effect of *TAp73α* overexpression on the metabolomics of GCs in a negative model.** (**A**) Volcano plot of metabolites with significant differences between control and *TAp73α* overexpression groups. The criteria for upregulated differential metabolites were VIP > 1, *p* < 0.05, and log2 (fold change) > 0, while the criteria for downregulated differential metabolites were VIP > 1, *p* < 0.05, and log2 (fold change) < 0. (**B**) Clustering heat map of differential metabolites between control and *TAp73α* overexpression groups. (**C**) Top 10 elevated and reduced differential metabolites identified between the control and *TAp73α* overexpression groups in the positive model. (**D**) KEGG pathways enriched by differential metabolites between control and *TAp73α* overexpression groups. The dashed line represents -log10 (*p*-value), corresponding to *p* < 0.05.

**Figure 6 ijms-26-06045-f006:**
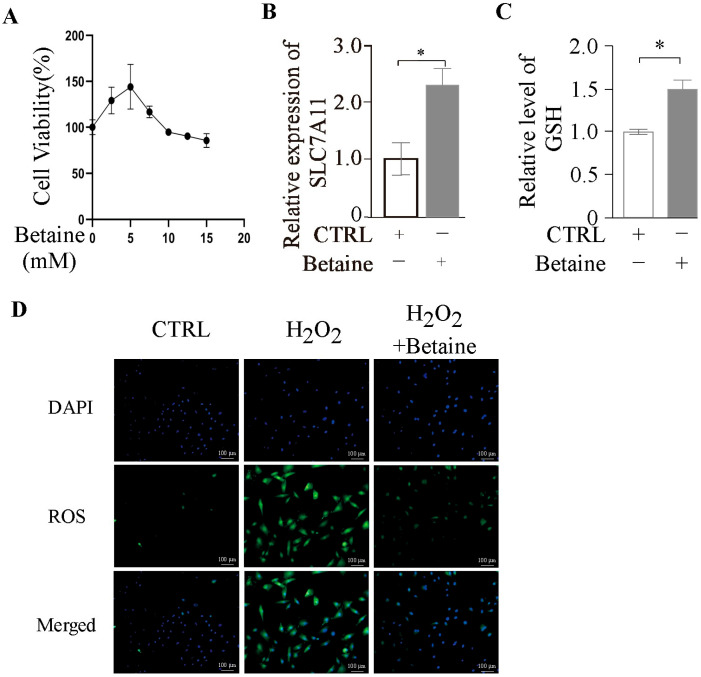
**Effect of betaine on GCs.** (**A**) Cell viability of GCs treated with different concentrations of betaine, as assessed by the CCK-8 assay. (**B**) Relative mRNA expression levels of *SLC7A11* in GCs after betaine treatment. (**C**) Intracellular GSH levels following betaine treatment. (**D**) Intracellular reactive oxygen species (ROS) levels measured using the DCFH-DA fluorescent probe in GCs treated with Erastin, betaine, or their combination. Data are presented as mean ± SD. * *p* < 0.05.

**Figure 7 ijms-26-06045-f007:**
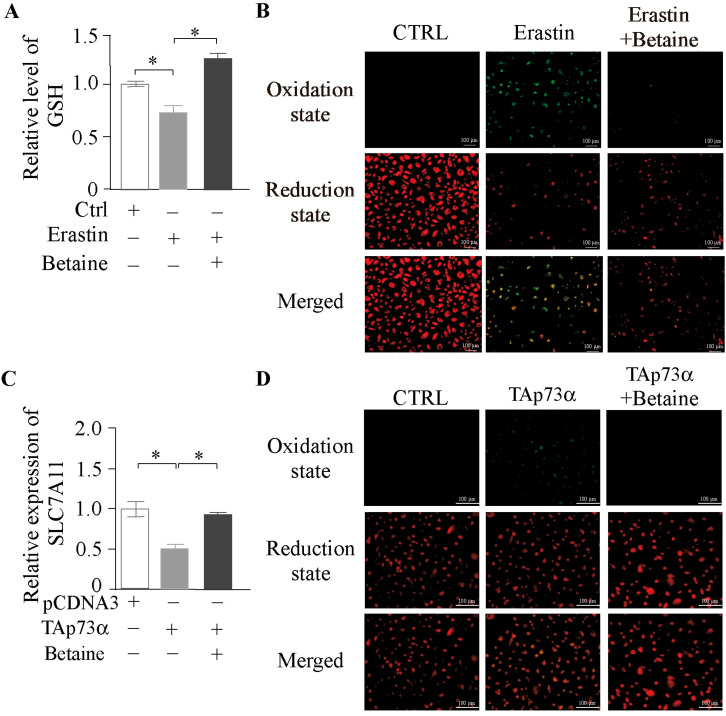
**Betaine attenuates *p73*-mediated ferroptosis in granulosa cells.** (**A**) GSH levels in GCs treated with Erastin, betaine, or their combination. (**B**) Lipid ROS level in GCs treated with Erastin, betaine, or their combination. (**C**) Relative mRNA expression levels of *SLC7A11* in GCs after *TAp73α* overexpression, betaine, or their combination. (**D**) Lipid ROS level in GCs treated with *TAp73α* overexpression, betaine, or their combination. Data are presented as mean ± SD. * *p* < 0.05.

**Figure 8 ijms-26-06045-f008:**
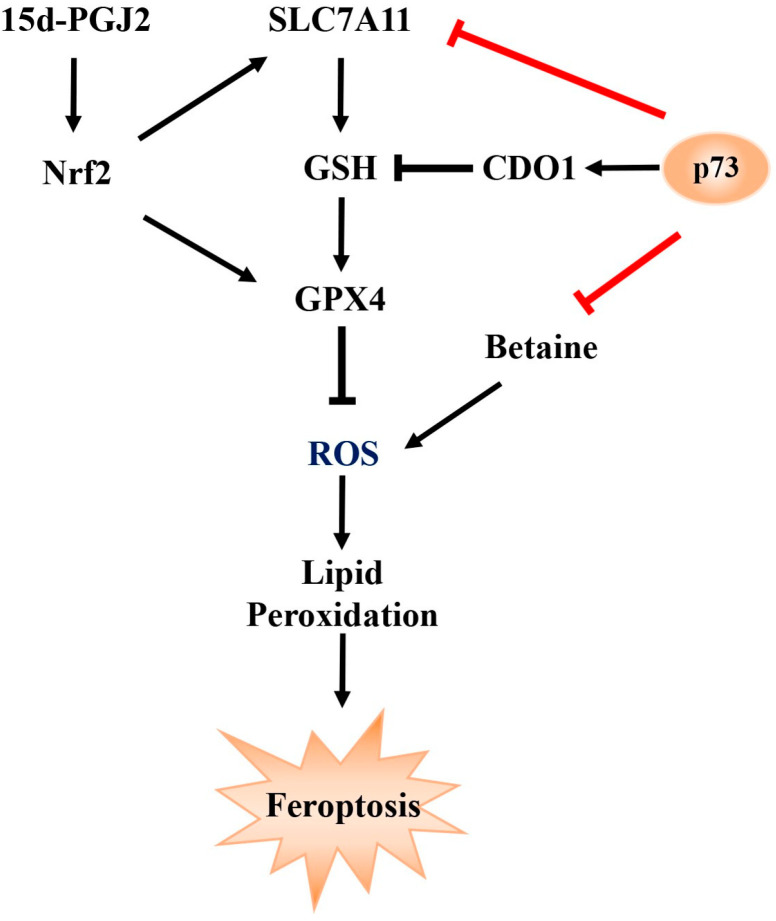
**Mechanistic pathways of *p73*-mediated ferroptosis involving SLC7A11 and betaine.** *p73* suppresses the transcriptional expression of SLC7A11 and reduces intracellular levels of betaine, a metabolite known to inhibit ROS accumulation. Arrowheads represent activation; blunt ends represent inhibition.

**Table 1 ijms-26-06045-t001:** All primer information.

Gene Name	Primer Sequence (5′-3′)
*ACTB*	GGTGCCCATCTATGAGGGGTACG
	TTCTCCTTGATGTCACGGACGATTTC
*GAPDH*	CACTTTGGCATCGTGGAGGGACTT
	AACAGACACGTTGGGAGTGGGGAC
*p73*	TGCCTGCTAACGGTGAGATGAACG
	GTCCCTGAGAGGTGAAGTACTCGATGC
*SLC7A11*	GTCCTGTCGCTGTTTGGAGCCTTGT
	CTGACACTCATGCTATTTAGGACCATCACC

## Data Availability

The data underlying this article will be shared upon reasonable request to the corresponding author.

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
