# Peer review of "Metabolomic Analysis Identifies Betaine as a Key Mediator of TAp73α-Induced Ferroptosis in Ovarian Granulosa Cells"

_ijms, 2025, doi:10.3390/ijms26136045_

Round 1
Reviewer 1 Report
Comments and Suggestions for Authors
The present paper examined the regulatory effect of TAp73alpha on ferroptosis and found the important role of betaine in a metabolic modulation of TAp73alpha-induced ferroptosis in ovarian granulosa cell. The studies were well designed and the analyses were solid and reliable. The manuscript is well written and readable. I think that this interesting work would certainly advance our understanding of the molecular mechanism of the TAp73alpha-induced ferroptosis. I recommend this paper for publication in the Journal. However, I raise some concerns that need to be addressed before publication. If those concerns are adequately addressed in the revised manuscript, this interesting report would be significantly strengthened.
Concerns that need to be addressed are:
[1] Please make an Abbreviation List on the first page or before References: Words in Abbreviations should be alphabetical order for readers to easily pick up words.
Please put the following and other words into the List. GC, GSH, HPG, GnRH, LH, FSH, PPP, ROS, SLC7A11, CCK, PFKL, G6PD, GLS-2, PGJ2, 12-OxoETE, COX, VTG, MDA, GPX and others, if they appear more than twice in the paper.
Tap73alpha, betaine and erastin might be also defined in the List. What are the chemical structures of betaine and erastin?
[2] Very importantly, it is essential to make a graphical figure as a new Figure 1 that visualizes prospects how proteins/substances/genes (including words of the Abbreviation List) play roles in physiological functions described in this paper.
Additional similar (but different) cartoon in the middle of 3. Discussion (rather than at 5. Conclusion) also would certainly be beneficial for general readers to easily grasp important points obtained in this paper. This cartoon should include PFKL, GLS-2 PGJ2, 12-OxETE, COS, VTG, MDA, GPX and others in addition to the above new Figure 1.
[3] References-1: Some papers (such as Refs. 3-9 and many others) describe both the first page and last pages. But some papers (such as Refs. 1, 2, 10-13 and many others) describe only first pages. Please remedy or normalize this issue.
References-2: Some journals are described in the abbreviated form (for example, Ref. 8, 29, 38, 39 and others), while other journals are described in full without abbreviations. Please remedy or normalize this issue.
Author Response
Reviewer #1:
The present paper examined the regulatory effect of TAp73alpha on ferroptosis and found the important role of betaine in a metabolic modulation of TAp73alpha-induced ferroptosis in ovarian granulosa cell. The studies were well designed and the analyses were solid and reliable. The manuscript is well written and readable. I think that this interesting work would certainly advance our understanding of the molecular mechanism of the TAp73alpha-induced ferroptosis. I recommend this paper for publication in the Journal. However, I raise some concerns that need to be addressed before publication. If those concerns are adequately addressed in the revised manuscript, this interesting report would be significantly strengthened.
Concerns that need to be addressed are:
[1] Please make an Abbreviation List on the first page or before References: Words in Abbreviations should be alphabetical order for readers to easily pick up words.
Please put the following and other words into the List. GC, GSH, HPG, GnRH, LH, FSH, PPP, ROS, SLC7A11, CCK, PFKL, G6PD, GLS-2, PGJ2, 12-OxoETE, COX, VTG, MDA, GPX and others, if they appear more than twice in the paper.
Tap73alpha, betaine and erastin might be also defined in the List. What are the chemical structures of betaine and erastin?
Response: Thank you for your valuable comments. We have now added the Abbreviation List in alphabetical order before the References section, as suggested. Please kindly refer to the updated manuscript for your review. Please kindly refer to Lines 447-462 in the revised manuscript for details. In addition, the chemical structures of betaine and erastin have been included in the revised version. Please refer to Line 79 for erastin and Lines 260–261 for betaine. We appreciate your helpful feedback, which has improved the clarity and completeness of our manuscript.
- Very importantly, it is essential to make a graphical figure as a new Figure 1 that visualizes prospects how proteins/substances/genes (including words of the Abbreviation List) play roles in physiological functions described in this paper.
Response: Thank you for your valuable suggestion. As per your recommendation, we have added a new Figure 1 to illustrate how key proteins, substances, and genes (including those listed in the Abbreviation List) participate in the physiological functions described in our study. This figure has been incorporated into the Introduction section, with the title: “A Conceptual Model of Ferroptosis Regulation Involving p73, SLC7A11, GPX4, GSH, ROS, and Nrf2.” Please kindly refer to Lines 68–80 in the revised manuscript for details.
[3] Additional similar (but different) cartoon in the middle of 3. Discussion (rather than at 5. Conclusion) also would certainly be beneficial for general readers to easily grasp important points obtained in this paper. This cartoon should include PFKL, GLS-2 PGJ2, 12-OxETE, COS, VTG, MDA, GPX and others in addition to the above new Figure 1.
Response: Thank you for your helpful suggestion. In response, we have created a new Figure 6, placed in the Discussion section, as recommended. This figure highlights two key findings of our study—namely, that p73 suppresses the transcriptional expression of SLC7A11 and reduces intracellular levels of betaine. Please kindly refer to Lines 298–303 in the revised manuscript.
[4] References-1: Some papers (such as Refs. 3-9 and many others) describe both the first page and last pages. But some papers (such as Refs. 1, 2, 10-13 and many others) describe only first pages. Please remedy or normalize this issue.
Response: Thank you for your comment. We have re-checked the references you mentioned. Indeed, the articles such as Refs. 1, 2, and 10–13 do not have traditional page ranges but are published with a single article number. Therefore, only one page number is shown. We have ensured that all references are formatted consistently based on the publication style of each journal.
References-2: Some journals are described in the abbreviated form (for example, Ref. 8, 29, 38, 39 and others), while other journals are described in full without abbreviations. Please remedy or normalize this issue.
Response: Thank you for your feedback on the journal name format. To ensure consistency, we have carefully reviewed all references and standardized the journal names to use the full titles throughout. All modifications have been highlighted in yellow in the revised manuscript for your convenience. Please kindly refer to the updated reference list.

Reviewer 2 Report
Comments and Suggestions for Authors
This study presents a novel intersection of transcriptional regulation (via TAp73α) cells death via ferroptosis and metabolic reprogramming (betaine) in ovarian granulosa cells, which has implications in reproductive biology. The methodology appears rigorous, and results are well supported. The manuscript is close to publication, with a few refinements suggested.
- L 75-76 Redefine this line.
- L 80, 106, 147, 188, remove empty lines in manuscript.
- L 344, The specification regarding the progenesis QI version is necessary for data processing transparency.
- L 360, Include kit manufacturer name and catalog number.
- The methodology and result sections would benefit from numbering the subheadings to improve clarity and navigation throughout the manuscript
- It is suggested to propose specific future directions that stem from the findings.
- The conclusion section closely mirrors the abstract, please redefine it.
- Moderate English revision is required in the manuscript.
Moderate English revision is required in the manuscript.
Author Response
Reviewer #2:
This study presents a novel intersection of transcriptional regulation (via TAp73α) cells death via ferroptosis and metabolic reprogramming (betaine) in ovarian granulosa cells, which has implications in reproductive biology. The methodology appears rigorous, and results are well supported. The manuscript is close to publication, with a few refinements suggested.
L 75-76 Redefine this line.
Response: Thank you for your valuable suggestion. We have redefined the sentence as requested. Please see the revised version in Lines 83-84 of the manuscript.
L 80, 106, 147, 188, remove empty lines in manuscript.
Response: We have carefully reviewed the manuscript and removed all empty lines at the specified locations (L 80, 106, 147, 188). The document is now formatted to maintain consistent spacing throughout.
L 344, The specification regarding the progenesis QI version is necessary for data processing transparency.
Response: Thank you for your comment. We have clarified the version of the software in the revised Methods section. The updated sentence now reads:
“Raw data files obtained from LC-MS/MS analysis were processed using Progenesis QI software (version 4.0).”
L 360, Include kit manufacturer name and catalog number.
Response: We have added the requested information to ensure full transparency. The revised text now includes:
GSH determination using the Micro Reduced Glutathione (GSH) Assay Kit (Manufacturer: Solarbio Life Sciences, Catalog Number: BC1175), following the manufacturer's instructions.
Lipid peroxidation content was determined using the Lipid Peroxidation Assay Kit with BODIPY 581/591 C11 (Manufacturer: Beyotime Institute of Biotechnology, Catalog Number: S0043S), following the manufacturer's instructions
ROS determination using the Reactive Oxygen Species Assay Kit (Manufacturer: Solarbio Life Sciences, Catalog Number: CA1410), following the manufacturer's instructions.
Cell Counting Kit-8 (CCK-8) Assay using the CCK-8 Kit (Manufacturer: Coolaber, Catalog Number: SK2060), following the manufacturer's instructions.
The methodology and result sections would benefit from numbering the subheadings to improve clarity and navigation throughout the manuscript
Response: We agree that numbered subheadings will enhance the readability and organization of the manuscript. We have restructured the Methodology and Results sections by implementing a hierarchical numbering system (eg., 2.1, 2.2, 4.1, 4.2) for all subheadings. This change improves navigation and cross-referencing within the document.
It is suggested to propose specific future directions that stem from the findings.
Response: Thank you for your helpful suggestion. We have added specific future directions based on our findings in the Conclusion section. Please see the revised text in Lines 418-423 of the manuscript.
The conclusion section closely mirrors the abstract, please redefine it.
Response: Thank you for your valuable comment. We have revised the Conclusion section to avoid repetition and better emphasize the key findings and implications of our study. Please refer to the updated content in Lines 414-423 of the manuscript.
Moderate English revision is required in the manuscript.
Response: Thank you for your suggestion. We have revised the manuscript with moderate English editing as requested. Please refer to the highlighted red text for the specific changes.
